# A New Policy of Water Resources and Environmental Regulation in China

**DOI:** 10.3390/ijerph20032556

**Published:** 2023-01-31

**Authors:** Peipei Zhang, Changbo Qin, Lei Yu, Liyan Yang, Lu Lu

**Affiliations:** Institute of Strategic Planning, Chinese Academy of Environmental Planning, Ministry of Ecology and Environment, Beijing 100012, China

**Keywords:** water resource and environment zoning, water environmental-quality standards, upper limit of water-resource utilization, permit list for water environment

## Abstract

As a developing country, China is facing serious water pollution and scarcity, which indicates the need for integrated water-resource and environmental regulations. Zoning policies have undergone significant advancements to enhance water-resource utilization in China. However, conflicts and overlaps still exist among these policies. To integrate these zoning policies and regulations, the “Three Lines One Permit” (TLOP) water-environment policy was formulated as a new framework, which included the goal for water quality, upper limits on water-resource utilization, and a permit list. This study presents the main achievements of the TLOP as a case-study in Jinan. The territories of Jinan were divided into 158 water-environment control-units (WECUs) and classified into two types of protected zones, three types of pollution-control zones, and ordinary zones. The total maximum pollutant-loads in the 158 WECUs, and 138 townships were calculated. The water-resource-utilization indicator values and ecological demand of key rivers were specified. The permit lists for the water environment at macroscale, mesoscale, and microscale were compiled from four perspectives: spatial constraints, emissions control, risk prevention, and resource utilization. Finally, suggestions were proposed to promote a more scientific and efficient TLOP policy to enhance human–water harmony.

## 1. Introduction

The rapid socioeconomic-development is outpacing the water carrying- and supplying-capacities in many water-stressed regions in China. As a rapidly developing country, China is also facing serious water crises [1,2], such as over exploitation and inefficient utilization of water resources, deteriorating water-quality and hydro-morphological degradation, which pose a great threat to human survival and sustainable development [3,4]. A variety of measures have been adopted to reduce pollution and improve water quality (WQ), such as the Ten-Point Water Plan issued in 2015 [5], but some issues remain unresolved.

Currently, the most effective means of environmental protection are environmental monitoring, supervision, cadres responsibility (two duties for one post), and law enforcement. However, the source control of pollution remains relatively inadequate. As the most important policy for source control in China, the environmental impact assessment (EIA) relies less on accountability, compared to those in developed countries. Enterprises are pursuing the approval of the EIA, but they forgo the responsibility of environmental protection and pollution prevention. Therefore, in 2017, China initiated the “Three Lines One Permit” (TLOP) as a source control policy prior to EIA to improve the environmental governance capacity [6], based on the environmental protection planning [7], functional zoning [8], and strategic environmental assessments [9]. TLOP is a comprehensive policy for ecology, water, air, soil, and other resources. Among these resources, water is the most important. The TLOP aims to solve the following problems:

First, the disordered spatial development and irrational processes in urbanization and industrialization exacerbate the deterioration of water bodies [10], and has previously been overlooked. Zoning aids the limitations in TLOP, as it divides an area into sub-areas based on similar characteristics for the implementation of appropriate environmental measures [11,12,13]. China has previously explored zoning for protected areas [13], water functions, water environmental functions, water ecological functions [14], and watershed functions [15]. However, conflicts and overlaps exist between the current zoning-types. The TLOP aims to evaluate the spatial heterogeneity of water eco-environment functions and problems, integrate all current zoning-types in one map, and identify the protected and critical-pollutant areas.

Second, water-resource utilization in current urban-water systems is inefficient [16,17], leading to pollutant discharge into the water. In some regions, even the basic ecological-flows in rivers, streams, and wetlands for ecological water maintenance cannot be ensured [18]. Water-stressed, inefficient, and insufficient ecological-flow often results in the depletion and deterioration of water bodies [19]. The Water Pollution Prevention and Control Action Plan released in 2015 [20] highlighted the importance of comprehensively promoting water-pollution prevention, water ecological protection, and water-resource management [21]. With the deepening institutional reform of the State Council in China that began in 2018, the pollution-control policies will be unified under the jurisdiction of the Ministry of Ecology and Environment (MEE). These policies will cover water-function zoning, groundwater quality, and marine environment, which are conducive to the coordination of zoning and management requirements. The TLOP is the first policy that tries to improve WQ with regards to water environment, resources, and ecology.

Third, isolated and segmented regulations are less effective for improving government efficiency and capacity [22,23]. The TLOP was designed to integrate all current regulations of every water-environment control unit to eliminate the inconsistent management- requirements, laws, regulations, planning, etc.

In this study, we aim to introduce the TLOP in China, exhibiting its use in Jinan and the policy implications, and try to solve the existing problems in water environment management.

## 2. Water-Resource and Environment-Zoning Policy in China

### 2.1. Planning

The three indicators of total water amount, water-utilization efficiency and allowable wastewater-discharge have had a very big effect on water-resource management, which can be seen in Figure 1a. The overexploitation of groundwater for the past 40 years is the main cause of subsidence in the northern cities of China. In turn, all 32 provinces have delineated groundwater no-exploitation and restricted-exploitation zones. In addition, no-discharge zones and restricted-discharge zones were delineated through potable-water-source protection planning. As shown in Figure 1b, some zones have overlapping relationships.

The National Five-Year Water Ecological Function Plan (NFWEFP) that covers all of China, has been iterated six times since its implementation in 1995. The 13th NFWEFP includes 10 first-level zones, 338 s-level zones and 1840 third-level zones. Its practical implications are as follows: (1) the planning scope gradually covers the entire territorial space, and the scale can be strategically changed from macro to micro; (2) protection of high-function waters and improvement of severely polluted waters can both be pursued; and (3) varying environmental zoning can further improve WQ. The comparison of current plans related to WQ is shown in Table 1.

### 2.2. Zoning

The water-function zoning (WFZ) and water-environment-function zoning (WEFZ) are the most important zoning types for water resources and environment. The comparison of WFZ and WEFZ is summarized in Table 2.

The WFZ was approved by the State Council in 2011. It covered 10 key watersheds, 2069 rivers, and 248 lakes in China. It included 3396 water-function first-class zones, of which 1450 utilization zones were further delineated into 2862 water-function second-class zones. The WFZ focuses on the development and utilization of water resources, such as water-intake and sand-excavation control, but ignores pollution-emission control.

The WEFZ was issued by the MEE in 2002. It included 12,876 water-environment functional zones, covering 10 major watersheds, 5737 rivers, and 980 lakes of China. WEFZ plays a significant role in improving the WQ. Compared to WFZ, WEFZ covers more water bodies with a weaker legal basis, but is more effective in WQ management. Both of them fail to achieve the coordination of land and water. However, the WQ goals of WFZ and WEFZ have generated some conflicts in some regions because of the different administrative departments and objectives (Table 2). For example, the WQ standards for landscape-recreational water zones in WFZ are class I, class II, or class III, while the WQ standards for landscape-recreational water zones in WEFZ are class III, class IV, or class V. Additionally, the monitoring indicators for WFZ and WEFZ are different.

### 2.3. Three Lines One Permit

The above zoning or planning types are conflicting, and there is poor interconnection between them, so that they are poorly disseminated to the concerned individuals. The TLOP in China is the first attempt to establish countrywide integrated-water-environmental-zoning based on the previous planning or zoning, and with a finer space scale. In additional, the WQ goals, pollution sources, required load-reduction, and environment permits for each water-environment control unit (WECU) was determined systematically, on one map. This study aims to introduce the framework of TLOP and its practices through a case study.

## 3. Methodology and Regulatory Framework

### 3.1. Framework of Three Lines One Permit in Water

The framework of TLOP in water comprises the bottom line for water-environmental-quality (BLWEQ), the upper limit for water-resource utilization (ULWRU), and the permit lists of the water environment (PLWE). The framework of TLOP in water mainly includes five steps, which are depicted in Figure 2.
(1)Comprehensive analysis: clarify all existing work foundations (such as related-research production, monitoring data, planning- or zoning-data, etc.), comprehensively identify the key problems of the water environment and resource, delimit the WECU according to the characteristics of the natural catchment and township boundary based on SWAT model or GIS [24], and identify the critical areas of these WECUs.(2)Zoning: conduct spatial overlay of the protected areas for potable water, important wetlands, fish migration routes, etc., and identify the areas that need to be protected; identify areas with high-intensity industrial pollution; analyze the spatiotemporal distribution of water pollutants from agricultural nonpoint-sources (NPS), domestic-sewage discharge, centralized treatment facilities, livestock and poultry, and the critical-source areas.(3)Phased goals of WQ: calculate the water environmental capacity; analyze the potential improvements of the WQ; formulate the WQ goals and calculate the total maximum annual-load (TMAL) permitted to be discharged into waters in 2020, 2025, 2030, and 2035.(4)ULWRU: analyze the utilization status of water resources, including the total amount, intensity, and efficiency of water-resource exploitation, aiming for WQ improvement and sustainable development.(5)PLWE: compile the permit list for water environments, based on the abovementioned steps.

### 3.2. Phased Goals of Water Quality

The WQ goals should be set in line with the following principles: (1) “only better, not worse” is the slogan of the nationwide battle to prevent and control pollution in China. For water-bodies in China that do not meet the applicable water-quality standards, it is required that the government should emphasize governance and reduce pollutant emissions, and promote the environmental-quality more effectively. If the applicable water-quality standards are reached, the pollutant discharge will be controlled within the environmental-capacity limitation and will not cause environmental pollution, loss of functional use or deterioration in environmental quality; (2) meet the requirements of WFZ and WEFZ before 2035; (3) the WQ goal in 2020 should exceed the requirements of the Water Pollution Action Plan (WPAP) and the target statement of responsibility for water-pollution prevention-and-control in each province and city; (4) the current and future economic, social, and management scenarios should be considered in the WQ goals. According to the guidelines of the TLOP [25], the WQ goals of all monitoring sections and WECUs should be set for the years 2020, 2025 and 2035. For achieving the WQ goals, the identification of pollution sources [26] and the computation of the required load-reduction [27] are equally important in restoring a water-body. Since the 1990s, the Chinese government has started changing the environmental-management strategy from solely pollutant-concentration control to a total maximum-daily-load program [28,29]. Calculating the total maximum annual-load (TMAL) includes five steps. First, the water environmental-carrying-capacity of each river and WECU needs to be calculated, in line with the WQ goals. Then, all the point-source pollutants from industries, poultry-breeding farms, and centralized sewage-treatment facilities need to be summarized for each WECU; in addition, the NPS pollutants from urban storm-runoff, agricultural planting, domestic-sewage direct discharge, and aquaculture need to be predicted and accounted for, for each WECU. Third, the load-reduction potential and increments in various pollutant sources under different socioeconomic developmental and environmental management scenarios need to be assessed. Fourth, a safety margin needs to be set [30]. Finally, the TMAL can be calculated using the following equation:(1)TMAL=∑WLA+∑LA+MOS
where WLA and LA indicate the allowable loads from point and nonpoint-source pollution, respectively, and MOS is the margin of safety. It is noteworthy that the spatially differentiated pollutant-reduction scheme and the WQ goal are bidirectionally optimized in the future.

*WLA* is the statistic of all the individual wastewater discharge based on the data from the second national survey of pollution sources in China. *LA* is calculated with the export coefficient model [31], which has been widely used to estimate nonpoint- source pollution-loads for its ease of application. The *MOS* is 10% in this study, which is typically assigned by making conservative assumptions or specified explicitly as a percentage (e.g., 5–10%) of the TMAL [32]. The TMAL map was classified into five classes using the Jenks natural breaks classification method with the help of ArcGIS 10.3. Map makers use the Jenks method to determine modest breaks in a data set by classifying similar values. Class-break values are determined, which best group similar values and maximize the differences among classes [33].

### 3.3. Zoning Criteria

The overlay analysis of spatial datasets for zoning was conducted in ArcGIS 10.3 software (Beijing Normal University, Beijing, China). WECUs were the smallest grid units. To integrate the natural attributes and management needs, the WECUs were delineated based on hydrological-response control units, monitoring sections, and township boundaries. The hydrological-response control unit was delineated using a digital-elevation model (DEM) with a 30 m resolution for the hydrological-analysis module.

According to the TLOP technical guidelines, specifications, and detailed documents released by the MEE, three types of zones were classified, based on WECUs: priority protection, critical source, and ordinary zones. The priority-protection and critical-source zones were further subdivided in Jinan. Despite all efforts for the provision of clean potable water [34], more than 200 million people in China still use unsafe water sources [34,35]. Thus, aside from the conventional potable-water-source protected Zones (WSPZ), water-conservation zones (WCZ) were added to strengthen the protection of water sources, where only activities with minimal ecological impact are permitted. WCZ serve as a buffer zone to ensure the effective protection of WSPZ. The critical-source zones [36] were further delineated, based on different pollution sources. The WECUs in Jinan were classified into six types, with the following criteria:(1)Water-source-protection zones (WSPZ). WSPZ include first-class and second-class protection zones for potable-water sources, the strong percolation zone of the 72 famous springs and the core area of the nature reserves, the habitat and migration channel for rare and endangered aquatic organisms, and important aquatic germplasm reserves.(2)Water-conservation zones (WCZ). WCZ include the prospective reserve zones for potable-water sources, wetland reserves, weak percolation zones of famous springs, experimental and buffer zones of nature reserves, high-quality waters, and the ecological buffer-zone of water-bodies.(3)Critical-source-zones of industrial pollution (CSZIP). CSZIP mainly include industrial parks, industrial-agglomeration regions, and other regions with high industrial-pollution emissions. Industrial parks or industrial-agglomeration regions undergo high-intensity economic activities and high resource-consumption [37], so they need to be critically managed.(4)Critical-source zones of domestic pollution (CSZDP). Aside from industrial pollution, domestic pollution emission has been the focus of total emissions control and the pollution permitting policies in recent years [38]. The treatment rate of rural domestic sewage wastewater in China was only 22% in 2018. The water contamination due to pollution from domestic wastewater, sewage treatment facilities, livestock, and poultry breeding sources are categorized as CSZDP. The pollution loads are calculated with empirical coefficient methods.(5)Critical-source zones of agricultural pollution (CSZAP). It is believed that NPS pollution threatens the regional water environment, wherein agricultural NPS pollution poses the greatest risk [39]. CSZAP can provide strong support for reducing NPS pollution and the pollution loads are identified by the average monthly/unit area pollution load [40] or load–area curve methods [41]. In this study, the average monthly/unit area pollution load was used.(6)Ordinary zones. The other regions not under the five mentioned categories are classified as ordinary zones.

### 3.4. Permit List of Water Environment for Human Activities

The permit list of water environment (PLWE) is a summary of current policies and new regulations from the BLWEQ and ULWRU. PLWE integrates all the regulations in regions, watersheds, and units based on spatial, pollution control, risk prevention, and resource utilization constraints.

### 3.5. Study Area and Data Sources

#### 3.5.1. Study Area

Jinan is located within Shandong province in China, within latitudes 36°32′–36°51′ N and longitudes 116°49′–117°14′ E, as shown in Figure 3. It is bordered by the Tai Mountain to the south and the Yellow River to the north. The southern region has a more rugged terrain than the northern portion. The altitude of Jinan ranges from 23 to 975 m above sea level [42]. Jinan has a semi-humid continental monsoon climate with an annual average temperature of 14 °C and annual average precipitation of 650–700 mm [43].

The population of Jinan increased from 3.19 million in 1952 to 8.70 million in 2018. The urbanized land increased from 24.6 km^2^ in 1949 to more than 479 km^2^ in 2014 [43]. In turn, Jinan’s ecological space has been severely affected, leading to the overexploitation of the environmental carrying-capacity and an increase in pollutant emissions. The urban-development patterns and industry structures are mismatched with the environmental conditions. Jinan was known as the “City of Springs” because of its 72 famous springs. However, due to the overexploitation of groundwater, the famous springs in eastern Jinan have been drying up for the past 20 years. The imbalance between the development and protection of spring water has aggravated these conditions [42]. 

#### 3.5.2. Data Sources

The data sources used in this study are shown in Table 3. First, the basic geographical information data, including DEM, land use, and river map, were used to refine WECUs, identify different land-use types, and predict NPS pollution-loads. Second, hydrological data from 2005 to 2016 and WQ data from 2012 to 2016 were combined. Finally, to fully link the requirements of existing management and control zones, various protected areas and plans were collected, to delineate WECUs and zones. 

## 4. Results

### 4.1. Bottom Line for Water Environmental-Quality

Firstly, nine national WECU were divided into 158 WECUs, and each protected area was maintained as an independent WECU for convenient management. The classified WQ goals of the 67 municipal monitoring sections and 158 WECUs in 2020 and 2035 were formulated in Jinan as shown in Figure 4. To improve the water quality, the long-term WQ target has a more stringent pollution-control standard.

The TMAL of each WECU for 2020 and 2035 (Figure 5) was calculated for subsequent EIA approval, issuance of discharge license, etc. The classes were shown with colors from beige to dark brown. Dark-brown and brown WECUs demonstrate proposed high-TMAL target areas, orange WECUs demonstrate medium-TMAL target areas and beige areas demonstrate low-TMAL target areas, in Figure 5.

Furthermore, to meet the need of Jinan’s Bureau of Ecology and Environment, the TMAL were reflected in 138 townships. Compared with the pollutant loads in 2016, the long-term target allowed for fewer contaminants to be present in the aquatic environment. Therefore, the reduction percentage was expected to increase as the remediation target changed from the short-term target to the long-term one. The load-reduction rate of COD, ammonia nitrogen (NH_4_-N), and total phosphorus (TP) from 2020 to 2035 increased from 8% to 20%, 35% to 45%, and 36% to 48%, respectively.

The water environmental-zoning is shown in Figure 6. The priority protected-zones included WSPZ and WSCZ, occupying 43.8% of the territory of Jinan, mainly distributed in the south of the Yellow River. The famous hot-spring reserves area in the southern mountainous region of Jinan was categorized as WSCZ. CSZIP were mainly concentrated north of Licheng and Zhangqiu districts, occupying 18.4% of Jinan. CSZDP were mainly distributed among the Tianqiao, Shizhong, and Huaiyin districts, occupying 35.3% of Jinan. The Shanghe County and northwestern Pingyin County were categorized as CSZAP, occupying 2% of Jinan. Other regions of Pingyin and Jiyang counties were ordinary zones.

The regulatory zones were classified into six types, including two protection types, three critical-regulatory types, and an ordinary-restriction type. WSPZ and WCZ represent areas that need to be protected and restored, while CSZIP, CSZDP, and CSZAP are critical areas where pollutant reduction measures need to be implemented. The water resource and environmental-zoning map of Jinan can be seen in Figure 6.

### 4.2. Upper-Limit Line for Water-Resource Use

To maintain water ecological function, 13 rivers were selected for their severe pollution or highly important roles, and their ecological water demand was estimated in accordance with the Standard for Calculating Water Requirement of River and Lake Eco-environment (SL/Z712-2014) [44]. The 13 rivers and underground-water zone of Jinan can be seen in Figure 7. The design of reclaimed-water-reuse programs for the nearby sewage-treatment plant was encouraged, to promote the ecological water supply. To protect the famous springs of Jinan, the strong-leakage zones, normal-leakage zones and cluster of springs from different plans were collected. The strong-leakage zone is the direct-recharge area of the Jinan springs, and the location of the zone is clearly defined, so that in extreme drought conditions the normal gushing of the springs can be ensured through artificial recharge and other measures. The normal-leakage zone has a lower potential for artificial recharge than the strong-leakage zone. In addition, the strong-leakage and normal-leakage zones of the famous springs in Figure 7 were also classified into WSPZ and WCZ in Figure 6, respectively.

The URWRU integrated the regulations for water resources, ecology, environment, surface water, and groundwater zonings. For example, in strong-leakage zones such as the famous springs, the previous requirement was “no more development land, and gradually restore the leak-out effect through low-impact development technology.” In the URWRU, high-pollution or high-risk projects (such as paper making, printing and dyeing, and petrochemical) are forbidden within WSPZ and WCZ.

### 4.3. Permit Lists of Water Environmental

In Jinan, PLWE encompasses three different spatial scales (macro, meso, and micro) from four aspects: spatial constraints, pollution control, risk prevention, and resource utilization.

The macroscale lists are the general requirements in different zones for providing a large-scale decision-making reference for both industrial and structural upgradation and spatial optimization, as shown in Table 4. The permit list for priority protected- zones focuses on the spatial restrictions of industrial development and urban growth to protect high-function water and clean water. The permit list for critical-source zones focuses on methods for reducing pollutant emissions.

The mesoscale lists are the specific management requirements for each WECU, which mainly include the attributes of WQ-monitoring sites, such as whether standards are being met, WQ goals for 2025 and 2035, major pollution-sources, priority- protection targets, and TMAL of COD, NH_4_-N, and TP.

The microscale lists include enterprises that need rectification owing to the intense consumption, pollution, and risks that they may pose to the implementation of key regulatory objectives in the near future.

PLWE combined various regulatory requirements for the first time, providing common regulations for the sub-zones, and differentiated regulations for each WECU, to improve the efficiency of eco-environmental management. Furthermore, the limited access and restricted requirements for each WECU were specified in the PLWE to retrieve data from the digital-information-management platform (DIMP), to achieve EIA approval and project site selection. The DIMP of TLOP was designed to connect with the interfaces of existing environmental-information platforms, such as EIA, emission-permit systems, and environmental monitoring.

## 5. Discussion and Implications

Integrated environmental-zoning is an interconnected approach to address the complexity of ecological and environmental problems, and it is regarded as a critical means of promoting sustainable development [45]. Existing studies have extended the theory of IEZ and proved that in Europe, Australia and the United States, IEZ can improve environmental performance better than traditional processes [6]. TLOP integrates all water-related regulations and establishes a countrywide water-environmental-zoning based on WECU, in China. The challenge involves coordinating the inherent conflicts and integrating different restrictions and goals into one blueprint without overlooking any information. Because of the discrepancy between the hydrological and administrative scales for implementing feasible management policies, traditional watershed-based management measures have limited application. Thus, to achieve the need for refined management of water environments within the period of the 14th Five-Year Plan, TLOP combined watershed-based and administration-based management through WECU, and fully considered the integrity of administrative divisions while maintaining the boundary integrity of small watersheds Townships are the smallest unit of land-use-planning in China, and studying the water environment at this unit level will make an important contribution to fine-scale management. Of course, some data necessary for these studies were difficult to obtain in China [34]. To promote the data sharing of TLOP and improve regulatory efficiency in the future, a DIMP was developed by the MEE.

Zoning can help the management of administrative regions and satisfy the requirements of the business application of administrative management. The zoning method in the TLOP can be applied in all the cities. For mountainous and hilly cities, the delineation of WECUs should put more emphasize on the impact of DEM. For cities in flat areas, the service scope of the sewage network has a greater impact on the delineation of WECUs. Additionally, the critical-source area of the pollution source can be divided into different types, based on various levels of pollution-emission intensity and characteristic in various cities. For example, for cities with abundant water resources in south China, aquaculture pollution may be serious, so critical source zone of aquaculture pollution should be set as a new type of zone. Considered the connectivity between surface water and groundwater, some famous-spring protection zones were grouped into WSPZ and WCZ in Jinan. The construction of a list of pollution sources is essential, but this study still has some limitations because of the poor foundational work and environmental data. Thus, dynamic updates and adjustments of goals should be considered every five years. Additionally, the TMAL need to be separated into seasons or months in the future. Uncertainty analysis was not carried out to select an MOS. Subjective or arbitrary specification of MOS might lead to overly conservative estimates and increased cost of implementation of pollution-control measures [46].

As a product of EIA reform, TLOP can be used to simplify the preparation and approval of the EIA report and may even replace parts of the EIA project. TLOP aids in decision-making on socioeconomic development. The zonings can also assist in project-site selection. The spatial regulatory zones with specific WQ goals could be a reliable basis for water-compensation policies [47], the spatial planning of territories [6], and five-year planning. The WQ goals and WECUs in TLOP can support the integration of WFZ and WEFZ, under the implementation of the institutional reform plan of the State Council in China. More applications should also be explored. It is worth mentioning that water ecology, such as water and soil loss and flood risk, are not discussed in this study, because these are included in the first line of the TLOP ecological-protection red line.

Undoubtedly, there are some insufficiencies in the TLOP policy. In regard to the lack of consideration of climate and biology variables in TLOP, studies are needed to develop watershed-zoning techniques capable of capturing tangible climatic and biological variabilities [45]. The eco-functional zoning of the aquatic ecosystem should be strengthened further in the TLOP [48]. Due to the complexity of the water-resource environment, a unified method on water environmental-risk zoning and ecological buffer zoning of rivers and lakes has not been established. ULWRU is relative weak regarding this policy, but not in Jinan, because the water resource in Jinan is vulnerable and scarce. That is, only if water resources have been the constraining factors affecting eco-social and social-ecological development and the transformation, would the ULWRU be strong and useful. If the water resource is rich, ULWRU would be weak, and would rely on the indicator limitation of the water consumption and utilization efficiency. More studies should be conducted to develop water-resource and environmental carrying-capacity optimization, and to adjust the economic layout and industry structure [49]. The TLOP has been introduced into the Yangtze River Protection Law, but its legal status still needs to be promoted. Subsequently, dynamic update, tracking, and evaluation measures should be designed to promote the application of TLOP in different departments. 

## 6. Conclusions

This study systematically summarized the zoning policy of the water environment and resources in China. To solve the problem of inherent conflicts, inconsistent external boundaries, and isolated and segmented regulations, China developed the TLOP policy. It is an integration of existing protection zoning, pollution-control zoning, function zoning, and resource-utilization policies. The main compilation process of TLOP is to identify problems, set WQ goals, delineate different zones, and compile access-lists. The main principles are problem orientated, goal orientated, and effect orientated. The framework and zoning approach of TLOP were described, and this can also be applied to other similar regions in China.

Taking Jinan as a case-study, this study systematically shows the main achievements of TLOP. The territorials of Jinan were delineated into 158 WECUs and six zone-types. The water-quality targets for 67 monitoring sections and 158 WECUs were formulated, and the total maximum pollutant-loads of 138 townships were specified. The water-resource-utilization indicator values and water ecological-demand were then specified. The macroscale, mesoscale, and microscale water-environment access-lists were compiled from four perspectives: spatial constraints, emissions control, risk prevention, and resource utilization.

The framework of TLOP and its insufficiencies were discussed. With the aid of DIMP, more dimensions of PLWE should be explored, such as the total maximum pollution-loads in the wet, normal, and dry seasons. Information dissemination should be strengthened, to promote TLOP. Ecological compensation, supervision and management, and the dynamic update, tracking, and evaluation of TLOP are also important, and must be urgently established.

## Figures and Tables

**Figure 1 ijerph-20-02556-f001:**
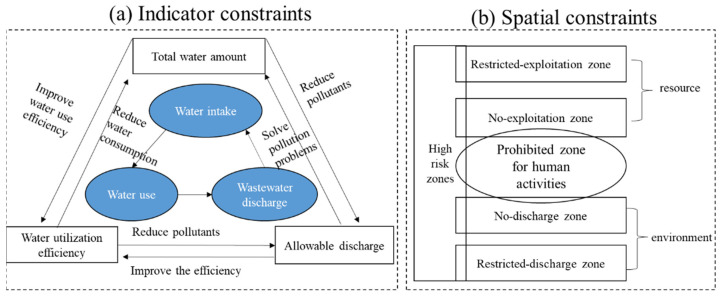
(**a**) Indicator- and (**b**) spatial-constraints of water management in China.

**Figure 2 ijerph-20-02556-f002:**
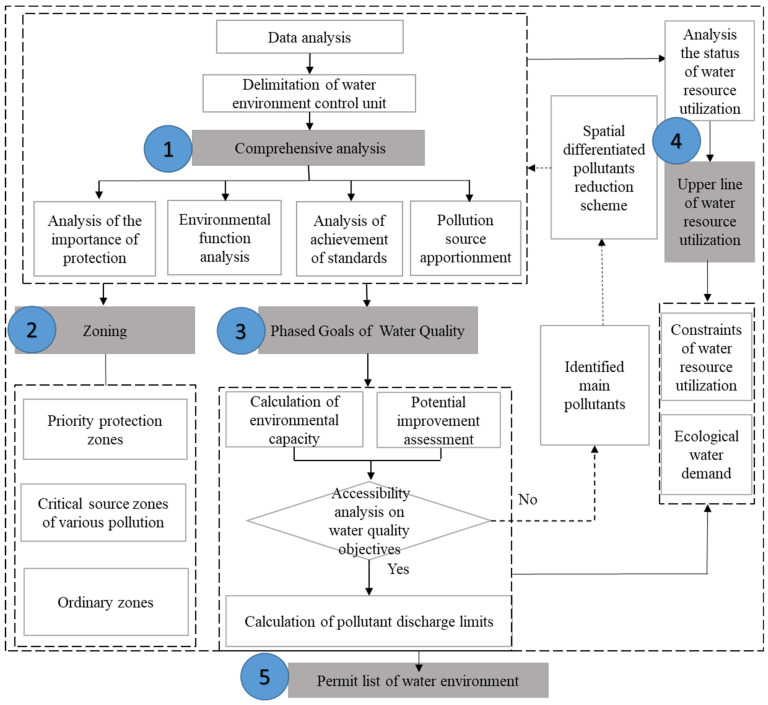
Framework of TOLP in water.

**Figure 3 ijerph-20-02556-f003:**
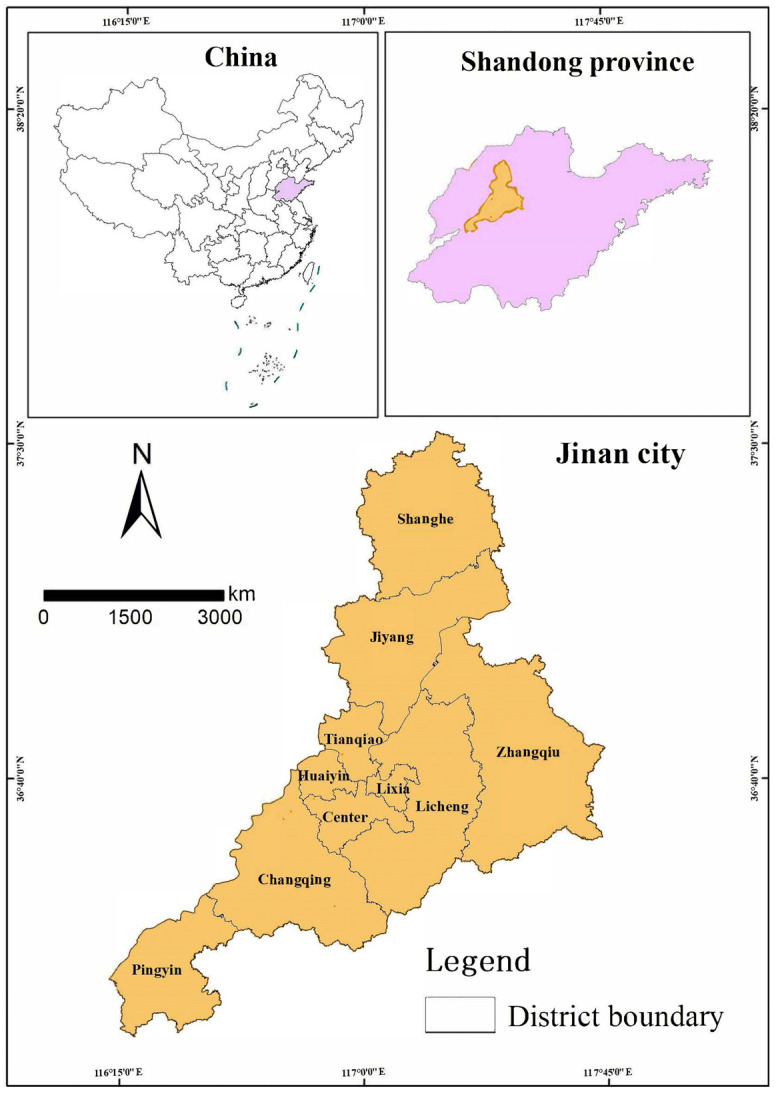
Location of Jinan City.

**Figure 4 ijerph-20-02556-f004:**
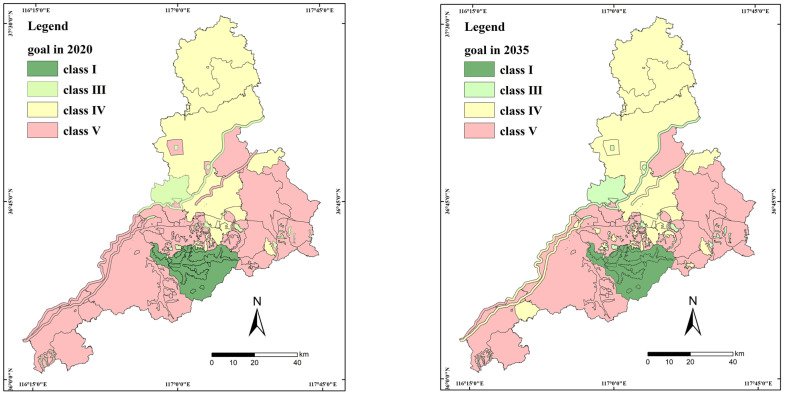
Classified WQ goals for Jinan in 2020 and 2035. The WQ classes for surface water were evaluated using environmental-quality standards for surface water (GB3838—2002) and the surface-water-environmental-quality assessment method (trial) in China. Class I: mainly for source of water and national nature-protection areas; Class II: mainly for Class I protection-areas for surface-water sources of centralized drinking-water, habitats for rare aquatic spawning grounds for fishes and shrimps as well as feeding grounds for young fishes; Class III: mainly for Class II protection-areas for surface-water sources of centralized drinking-water, wintering ground for fishes and shrimps, migration pathways, aquiculture areas as well as other fisheries areas and swimming area; Class IV: mainly for general industrial-water areas and entertainment water-areas not directly touched by humans; Class V: mainly for farmland water-areas and water areas for general landscape requirements.

**Figure 5 ijerph-20-02556-f005:**
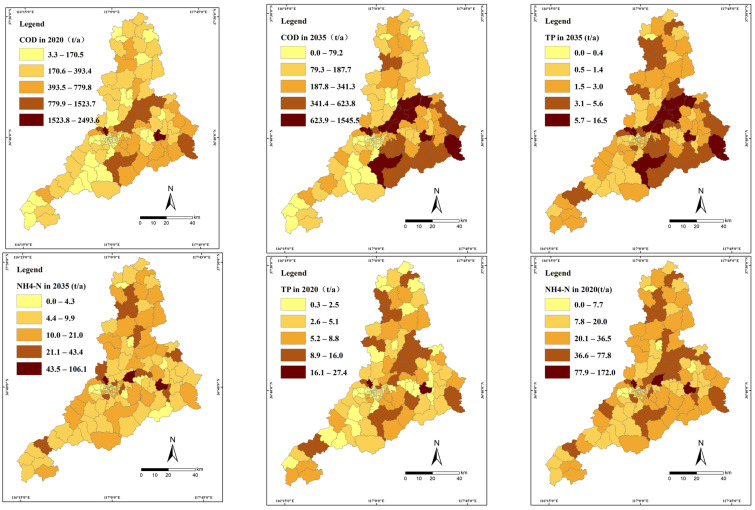
TMAL for COD, NH_4_-N, and TP in Jinan in 2020 and 2035.

**Figure 6 ijerph-20-02556-f006:**
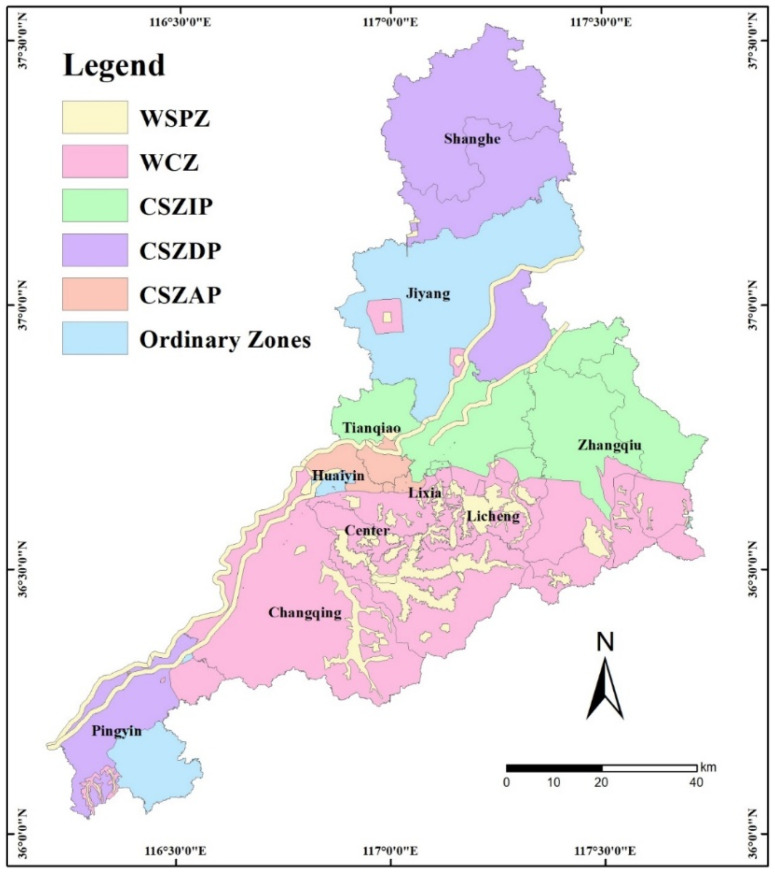
Water-resource and environmental-zoning map of Jinan. (Note: WSPZ is the acronym for water-source-protection zones; WCZ is the acronym for water-conservation zones; CSZIP is the acronym for critical-source zones of industrial pollution; CSZDP is the acronym for critical-source zones of domestic pollution; CSZAP is the acronym for critical-source zones of agricultural pollution.)

**Figure 7 ijerph-20-02556-f007:**
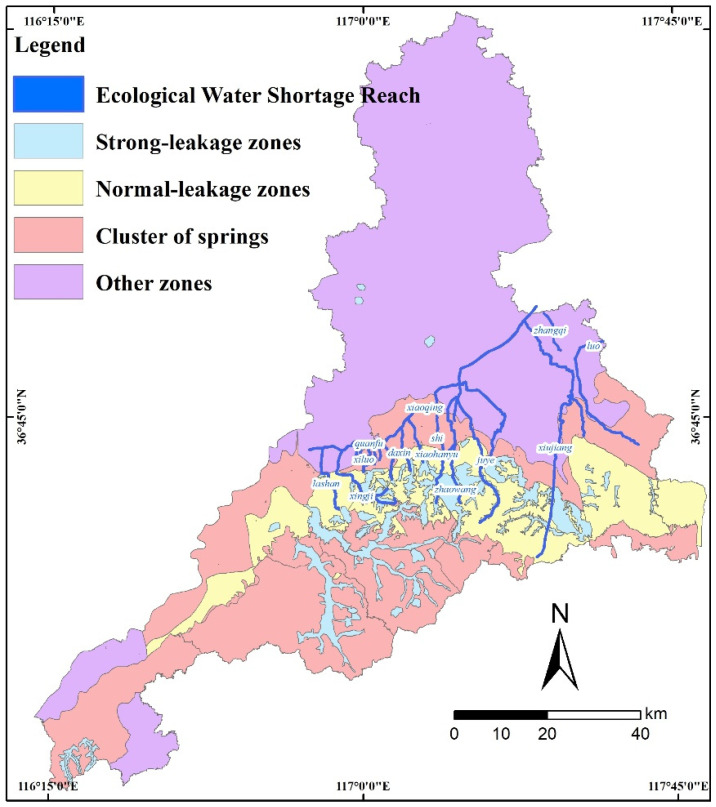
Ecological-water-shortage-reach and underground-water-zoning map of Jinan.

**Table 1 ijerph-20-02556-t001:** Current plans related to WQ.

Department	Plan	Scope ^1^	Zones	Main Restrictions
MWR	Groundwater exploitation and utilization planning	Not fully covered	No-exploitation zones	Ban on the exploitation of water resources
			Restricted-exploitation zones	Limit the exploitation scale and number of water resources
MEE	Potable-water-protection planning	Not fully covered	Primary reserves	No discharge of wastewater
			Secondary reserves	No discharge of industrial wastewater
			Prospective reserves	Limit the total amount of wastewater discharge
	NFWEFP	Fully covered	WQ-improvement zones	Reduce pollution
			Water ecological-maintenance zones	Safeguard the health of the ecosystem
			High-risk zones	Prevent and resolve environmental risks

^1^ Scope means the national territory of the planning covered.

**Table 2 ijerph-20-02556-t002:** Comparison of WFZ and WEFZ.

Type	Administrative Department	Legal Basis	Zones	Objective
WFZ	Former MWR	Water law of the People’s Republic of China	Rivers were classified into reserves, set-aside zones, utilization zones (potable-water zones, industrial zones, agricultural zones, fishery zones, landscape-recreation zones, transition zones, and pollution-control zones), and buffer zones.	Protect the functions of water bodies.
WEFZ	MEE	--	WQ ^1^ goals of rivers are classified into class I, class II, class III, class IV, and class V.	Protect the WQ of water bodies.

^1^ WQ means the water-quality standards that a specific river-segment needs to meet.

**Table 3 ijerph-20-02556-t003:** Data sources for the TLOP in Jinan.

Id	Types	Description	Source
1	DEM	DEM at 30 m resolution.	Geospatial data cloud
2	Land-use data	Land uses in 2017 in Jinan.	Jinan Land and Resources Bureau
3	River data	Thirty-three rivers in Jinan.	Jinan Water Conservancy Bureau
4	Annual Hydrological data	Hydrological data from 2005 to 2016.	Jinan Hydrological Yearbook
5	WQ data	Jinan Environmental Quality Report from 2012 to 2016.	Jinan Ecological Environment Bureau
6	Pollutant-emission data	Industry, sewage-treatment facilities, and livestock- and poultry-breeding sources.	Environmental Statistics of Jinan City in 2017
7	Discharge coefficient	Discharge coefficient of industrial and domestic sewage, and livestock- and poultry-breeding emissions.	National Pollution Source Census Office
8	Protection areas	Potable-water source protection-area, wetland reserve, etc.	Jinan Ecological Environment Bureau
9	Planning	Famous-springs protection planning.	Office of Famous Springs Protection in Jinan

**Table 4 ijerph-20-02556-t004:** PLWE of Jinan in macroscale.

Zones	Sub-Zones	Main Environmental-Management Measures
Priority protected zone	WSPZ	The WQ should reach category III or above. No new sewage-outlets. Dismantle or close all new sewage-discharge construction-projects unrelated to the protection of water sources and springs.
	WCZ	The WQ should reach class IV or above. TMAL should be decreased gradually. No high-pollution or high-risk industries, such as petrochemical, chemical, hazardous waste, electroplating, medicine, fertilizer, paper-making, chemicals, and lead-acid battery industries. No new concentrated livestock and poultry farms. Damaged mountain bodies should be restored and harnessed for afforestation.
Critical-source zones for pollution	CSZIP	Eliminate class-V water quality before 2030. The TMAL should be decreased gradually. Accelerate the transformation and upgradation of industrial structure, and the industrial projects must include industrial parks equipped with rain- and sewage-diversion pipelines.
	CSZDP	Eliminate class-V water quality before 2035. Improve the collection- and treatment-rate of domestic sewage. Construct rainfall- and sewage-diversion and pipe networks in urban built-up areas, and 75% of the initial rainwater should be treated.
	CSZAP	Eliminate class-V water quality before 2035. Concentrated livestock and poultry farms should construct and operate treatment facilities, and small-scale farms are encouraged to apply the ecological-cycle development mode. Extend formula fertilization after soil testing. Optimize agricultural production and planting structures.
Ordinary zone		Comply with conventional environmental-management regulations.

## Data Availability

Not applicable.

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
