# Peer review of "A New Policy of Water Resources and Environmental Regulation in China"

_ijerph, 2023, doi:10.3390/ijerph20032556_

Round 1
Reviewer 1 Report
Overall a useful and important paper - however many improvements are recommended. Please see the attached document for detail.

Reviewer 2 Report
This paper proposes a new environmental and resource management and control method "Three Lines One Permit" (TLOP). This method calculates the total pollution load by dividing the water environment control units, and then compiles the allowable discharge list of effluent environment through the water resource utilization index value and ecological demand. Finally, the permit lists of the water environment at macroscale, mesoscale, and microscale were compiled. This provides a new idea for water environment protection. Therefore, my opinion about this paper is positive. I would like to mention. However, there are still some minor problems in the article that need to be corrected. It is recommended to accept them after minor repairs.
Q1: in section 5 “Discussion and implications”, I feel ULWRU is relative weak in this policy, so it is recommended to be further discussed why or how to strengthen.
Q2: in section 5 “Discussion and implications”, as a new policy, whether the method of TLOP can be used in different city? According to the characteristics of different cities, which parts need to be deal with differently? More discussions on this question are appreciated.
Q3: Whether there are similar policy abroad, if so, it is suggested to add some comparative discussions.
Some other detailed comments are as follows:
Q4: “As a rapidly developing country, China is also facing serious water pollution crises (Gleick, 2009; Han, 2016), which pose a great threat to human survival and sustainable development (Chen et al., 2016; Liu et al., 2017).” It seems that the sentence is incomplete. It is suggested to add what serious water pollution has occurred. Page 1 Lines 27
Q5: “A variety of measures have been adopted to reduce pollution and improve water quality (WQ) (Gao et al., 2005), but some issues remain unresolved.” The sentence seems incomplete. It is suggested to add what measures have been implemented. Page 1 Lines 30
Q6: “The TLOP is the first policy that clearly addresses WQ goals from all aspects of water environment, resources, and ecology”. It is suggested to revised to “The TLOP is the first policy that try to improve WQ from water environment, resources, and ecology”, because the effect of this new policy haven’t been seen in this paper. Page 2 Lines 62-63.
Q7: “Section 2 summarizes the progress of water resource and environment zoning policies in China. Section 3 proposes the conceptual framework of TLOP and the technical system. Section 4 describes the main achievements of TLOP via a case study in Jinan. Section 5 discusses the innovations, insufficiencies, and policy implications. Finally, Section 6 summarizes the main findings and provides concluding remarks.” This paragraph is repeated, and it is suggested to delete or generalize that this study solves existing problems in water environment management through the case of Jinan. Page 2 Lines 76
Round 2
Reviewer 1 Report
Considerable attention has been given to address review comments. This appears well done so I recommend for publication